# Short- and Long-Term Effects of Underemployment on Workers’ Health: Empirical Analysis from the China Labor Force Dynamics Survey

**DOI:** 10.3390/ijerph192416695

**Published:** 2022-12-12

**Authors:** Nan Li, Huanhuan Liang, Yi Gao, Dan Wu

**Affiliations:** 1School of Business, Hubei University, Wuhan 430062, China; 2School of Management, Guilin University of Aerospace Technology, Guilin 541004, China; 3School of Public Administration, Zhongnan University of Economics and Law, Wuhan 430073, China; 4School of Economics and Management, Hainan Normal University, Haikou 571127, China

**Keywords:** underemployment, workers’ health, short-term effects, long-term effects

## Abstract

Underemployment is a global problem. This study aimed to assess the short- and long-term effects of underemployment (hidden unemployment) on workers’ health, using data from the China Labor-force Dynamic Survey (CLDS) 2016 and 2014. Indicators reflecting workers’ self-rated health, mental health, prevalence of illness over time, and employment status were analyzed using logit regression models, propensity score matching methods, and instrumental variable methods. Empirical analyses showed that: (1) In the short-term, the impact on health is multidimensional, with underemployment significantly associated with a decline in workers’ self-rated health, an increase in the propensity for depression, and an increase in the prevalence of illness over a certain period of time. (2) In the long-term, the experience of underemployment two years in the past is associated with a current decline in workers’ mental health. That is, the negative effects of underemployment on workers’ mental health persist and do not disappear rapidly over time. The results demonstrated that underemployment is detrimental to workers’ health in the short- and long-term. In the context of epidemic prevention and control, the government and society should focus on this expanding group, establish labor protection mechanisms, and reduce the multiple effects of underemployment on workers’ health.

## 1. Introduction

Full employment contributes to the well-being of workers, and is not only related to income, but also the physical and mental health of people. However, the recent spread of COVID-19 has hugely impacted the labor market. Employment status is divided into employment, unemployment, and hidden unemployment, that is, underemployment, which is the state of being “employed but not ideally employed” [1]. In many countries, increasing employment is an important factor behind economic growth. But an increase in jobs does not mean an increase in the quality of work. In reality, there has been an increase in informal work and more workers find themselves in jobs that do not pay enough to lift them and their families out of poverty [2]. Thus, in many countries, it is not unemployment that is the problem, but rather the existence of a large number of “underemployed” persons with no prospects for development. Underemployed workers have been increasingly normalized and are deemed as individuals who work less than 35 h/week but wish to work longer [3]. Although the problem of underemployment is not statistically significant, it is not uncommon in real life. In fact, a large proportion of the world’s workforce is underemployed [4]. In the event of major economic shocks, underemployed workers are forced to accept shorter work schedules to avoid long periods of unemployment [5]. It is believed that the problem of underemployment may worsen in China due to the influence of COVID-19 control measures and the global economic environment. However, statistics from labor force surveys around the world reveal that people often focus on the impact of explicit unemployment on workers’ health while neglecting that of underemployment (hidden unemployment). Consequently, does underemployment affect workers’ health? If yes, then what is the extent of its impact? This paper employs empirical data to explore the impact of underemployment on workers’ health and uses the evidence to provide effective support for the development of targeted intervention programs.

### 1.1. Health as a Commodity: A Theoretical Analysis

Mushkin proposed that “health is an investment”, a calculus which determines the amount of time people spend on market or non-market productive activities, and their effectiveness per unit of time [6]. It has been suggested that, given a certain level of socioeconomic development and other external conditions, an individual’s investment in health largely influences their health condition. The investment in health mainly comprises two aspects: economic investment (e.g., improving quality of life) and time investment (e.g., engaging in physical activity). Since each day has a fixed number of hours, an increase in the amount of time a worker spends doing one activity during the day will inevitably bring about a decrease in the time invested in other areas [7]. Since underemployed people spend less time working, they have relatively more time to invest on health and for physical activities [8]. Therefore, they are likely to be in better health condition. Overall, short periods of work not only help workers to earn an income, but also meet their psychosocial needs [9,10], while the energy and stamina expended in short periods of work are offset by the positive emotions, and self-affirmation experienced during the working hours [11]. In other words, the time effect caused by underemployment can enhance workers’ health.

Meanwhile, Grossman argued that “health is a consumer good” which frees people from the suffering caused by disease or illness, and gives them a sense of satisfaction and utility. He also proposed the first theoretical model for analyzing the demand for health. The model argues that people can purchase health in a monetary or nonmonetary manner [12]. Since underemployment is a kind of hidden unemployment, long-term underemployed workers usually do not have guaranteed income, which constrains their health care and other general expenditures, impacting their overall health condition [13,14]. Only workers with a certain degree of financial security are likely to invest in health, including stockpiling health knowledge, joining health clubs, purchasing exercise equipment, and having regular medical check-ups [15,16]. Moreover, low income can also cause psychological burdens or stress [17], which increases the depreciation rate of their health capital. Therefore, under the given external conditions, the incremental health capital of the underemployed is lower than that of fully employed workers. That is, the economic effects caused by underemployment can be detrimental to workers’ health.

Since underemployment is associated with more leisure time and possibly lower income, this paper argues that the time effect caused by underemployment has a positive impact, while the economic effect has a negative impact on workers’ health. The overall effect of underemployment is the combination of its time and economic effects (Figure 1).

### 1.2. Underemployment and Health: A Literature Review

While several studies have examined the effects of unemployment and overwork on workers’ health, only a few have assessed the relationship between underemployment and workers’ mental health using cross-sectional data. In fact, there are even fewer studies that have confirmed the negative effect of underemployment on the mental health of individuals [18,19]. Although many scholars have emphasized the importance of employment quality, the issue remains controversial. For example, Jahoda suggested that even a bad job is better than unemployment and argued that there is no association between underemployment and the health of workers [20]. However, Wu suggested that short-time jobs, though technically not underemployment, adversely affect health because they do not meet the psychosocial and economic needs of workers [21]. Underemployment can be described as a potential social stressor that pressurizes workers and may endanger their health. According to the Effort-Reward Imbalance (ERI) model, work can lead to poor mental health if people are not appropriately “satisfied” or “rewarded” [22]. Evidence also suggests that any change in role or environment that objectively requires adaptation causes a specific stress response which accumulates over time and causes varying degrees of damage to health [23,24]. Every employee is exposed to different social stressors, but they cope with the stress differently. For example, a study of East Asian immigrants and Vancouver residents found that underemployment was associated with higher levels of depression among Vancouver residents, while the same was not true for East Asian immigrants [25]. Thus, underemployment affects workers’ health, but the effects are neither as strong nor as consistent as is commonly believed.

In summary, there are several research lacunae to be filled. First, most studies on employment status and health simply distinguish between employment and unemployment without considering underemployment. This conceals the complexity of the relationship between work and health. Second, most studies on the impact of underemployment on workers’ health have focused on a single disease or health indicator and lack multidimensional in-depth analysis. This may greatly underestimate or overestimate the impact of underemployment on health. Therefore, this study discusses the relationship between underemployment and a series of physical and mental health indicators. Furthermore, as existing studies lack longitudinal analysis, this paper uses tracking data to analyze whether underemployment affects workers’ health in the long term.

## 2. Materials and Methods

### 2.1. Data Sources

In this study, we analyze the short- and long-term effects of underemployment on workers’ health, using the latest available data from the China Labor-force Dynamic Survey (CLDS) 2016 and 2014, published by the Social Science Research Center of Sun Yat-sen University. Although the data itself is not from the pandemic period, the findings of this paper may can be used as a reference for studying the impact of underemployment on workers’ health during this time, because stable employment has become increasingly constrained by the impact of COVID-19, and underemployment remains a continuing problem. Given this economic environment, particular attention must be paid to the health consequences of underemployment. CLDS is a tracking database, which integrates labor force, households, and communities. Probability sampling methods were used at multiple stages and levels proportional to the size of the labor force to determine the sample. This database is the first in China to adopt a rotating sample tracking method, one which suitably adapts to the drastic changes in China while accounting for the characteristics of the cross-sectional surveys. The sample covers 29 provinces, cities, and autonomous regions in China, except Hong Kong, Macau, Taiwan, Tibet, and Hainan, and is nationally representative. The CLDS 2016 was used as the base data to analyze the short-term impact of underemployment on workers’ health, while the long-term impact was estimated from CLDS 2014 underemployment data. The sample data were screened according to the study’s research question. First, the population outside the labor market was excluded, that is, students, homemakers, retired people, and people who have never worked. Second, the age of the labor force was limited to the legal working age population (16–60 years for men and 16–55 years for women). Third, extreme values and outliers were removed. After screening, the CLDS 2016 retained 10,563 observations, the CLDS 2014 retained 9703 observations, and the combined sample retained 4713 observations.

The Regulations of the State Council on the Working Hours of Employees and the Labor Law of the People’s Republic of China stipulate that workers should not work more than eight hours per day and 44 h per week on an average. Therefore, this paper considers weekly working hours greater than 44 h as excessive employment.

### 2.2. Model Setting and Variable Selection

First, we used logit models to analyze the short- and long-term effects of underemployment on workers’ health.
$$Health_{ip}=\alpha_{0}+\alpha_{1}Underemployment_{ip}+\alpha_{2}X_{i}+\beta_{p}+\varepsilon_{ip}$$

In this model, $Health_{ip}$ is the explanatory variable indicating the health status of individual worker *i* in province *p*, a dichotomous variable with healthy = 0, and unhealthy = 1. $Underemployment_{ip}$ indicates whether individual worker *i* in province *p* is underemployed, $X_{i}$ is the variable of individual characteristics, economic status, living habits, work characteristics, and insurance participation. $\beta_{p}$ indicates provincial and municipal fixed effects to address possible omitted variables by controlling for fixed effects, and $\varepsilon_{ip}$ is the random disturbance term.

Second, the baseline regression model was re-estimated using propensity score matching and the instrumental variables method to accurately assess the effect of underemployment on workers’ health by accounting for endogeneity issues such as possible omitted variables, reverse causality, and measurement errors in the previous model.

Propensity Score Matching [26]: Underemployment is selective, meaning, an individual’s age, education background, and marital status may affect the likelihood of being underemployed. These confounding variables contribute to the selectivity of the effect of underemployment on workers’ health. Therefore, it is important to consider this self-selectivity problem to correctly estimate the effect of underemployment on workers’ health. This paper introduced propensity score matching (PSM) analysis to further modify and test the baseline regression model.

Instrumental variable method [27]: The bidirectional causal relationship between underemployment and workers’ health status may lead to endogeneity problems, that is, the poorer the health status, the greater the likelihood of people experiencing underemployment. Since wages are an important reflection of labor costs, employers are likely to change the ratio of inputs of each factor of production to maximize benefits when workers’ wages change [28]. Thus, an increase in the minimum wage may lead employers to replace workers’ working hours with the amount of labor and capital. This in turn leads to an increase in the likelihood of underemployment. Consequently, the minimum wage in each city in 2016 was chosen as the instrumental variable to address the endogeneity issue in this study.

#### 2.2.1. Dependent Variables

Health status of workers was used as the dependent variable. In line with previous studies, the measure of workers’ health includes three dimensions: self-rated health, psychological health (presence of a tendency toward depression), and prevalence of illness over a certain period of time.

Self-rated health is an individual’s overall feeling and evaluation of various aspects of their health status. In the CLDS questionnaire, self-rated health is an ordered discrete variable corresponding to the question: “How healthy do you think you are now?”; the options for this question include: “very healthy, healthy, fair, relatively unhealthy, very unhealthy”. Self-rated health was dichotomized for data processing, with “very unhealthy” and “relatively unhealthy” defined as unhealthy and assigned a value of 1, and “fair”, “healthy” and “very healthy” defined as healthy and assigned a value of 0.

Depressive tendencies are often seen as an important indicator of workers’ mental health, which can be determined by scoring different workers on their depressive tendencies. In the CLDS questionnaire, depressive tendencies are measured using the CES-D-20 maturity scale, which corresponds to the question in the questionnaire: “In the past week, select the frequency of the following situations that occurred to you?” The four response options were assigned a value of 0, 1, 2 and 3, respectively, in reference to Radloff’s study [29], which defined a total score of 16 or more as having a tendency for depression and assigned a value of 1. Those with a total score of less than 16 were defined as not having depressive tendencies and assigned a value of 0.

Grossman (1972) suggested that health could be described by the time spent free from disease over a given period of time, and the Chinese health statistical yearbooks use disease prevalence as the main indicator of health status. In the CLDS questionnaire, the absence of illness during a given period is a dichotomous variable, and the corresponding question is: “Have you had any illness or injury in the past two weeks? In the data processing, the answer “no” is defined as no illness during a certain period of time and is assigned a value of 0, while the answer “yes” is defined as sickness within a certain period of time and is assigned a value of 1.

#### 2.2.2. Core Variables

Previous studies have defined underemployment as a preference, with fewer than 35 h of work per week and a desire for more working hours. The present study focuses more on utility, that is, workers’ dissatisfaction with their current working hours, which corresponds with a reduced sense of utility, which is more likely to affect workers’ health in terms of psychological satisfaction. This is in line with Otterbach and others, who defined underemployment as actually working fewer hours than preferred [30]. The questionnaire includes the following items: “How many hours a week do you typically work at your current or most recent job?” and “Please rate your current/last job status and whether you are satisfied with the hours you work.” With rising income and improvements in overall social welfare, some people prefer working at leisure more satisfying; however, voluntary short-time workers constitute a minority. For most people, underemployment is not a “good job” one to which they aspire, but rather an inferior job that they have no choice but to do. Therefore, an average working week of between 0 and 35 h and dissatisfaction with the number of working hours is defined as underemployment. This was assigned the value of 1. Other types of work time (not underemployment) are assigned the value of 0.

Since workers’ preferences and actual working hours are not always equal, some people actually work more hours than desired (over employment), some people work the same hours as desired (full employment), and some people work fewer hours than desired (underemployment). Therefore, referring to previous studies, this paper defines the average weekly working hours as between 35 and 44 h as fully employed and the average weekly working hours greater than 44 h as over employed. Based on this, the entire sample was divided into the following three categories: underemployed, fully employed, and over employed. Differences in their health condition were obtained by comparing the categories.

#### 2.2.3. Control Variables

Workers’ characteristics, economic status, living habits, work characteristics, and insurance participation were used as control variables. (1) Individual characteristics included gender, age, education level, household registration, marital status, and appearance. Gender facilitates the differentiation between people’s health conditions [31]; marriage helps people develop healthy lifestyles [32]; well-educated workers have better access to health care knowledge, higher levels of accessibility, and utilization of health care resources and services, and have relatively healthier lifestyles [33]; and health changes with age correspond with physiological processes that are difficult to control [34]. The household registration system is a social management system. With its increasingly obvious role in the labor market, we argue that the effect of underemployment on workers’ health may be influenced by individual characteristics. (2) Economic status includes personal income, family income, and housing. The economic status of workers can impact workers’ health by affecting their quality of life, nutrition level, lifestyle, and so on. In general, workers with higher income have better health [35,36]. (3) Lifestyle habits (e.g., smoking, alcohol consumption, exercise, etc.) are distributed differently across social groups; thus, this paper includes lifestyle variables in the analyses. (4) In terms of work characteristics, a good working environment reduces the chances of workers suffering from injuries, while transitory employment groups are often disadvantaged in the labor market [37]. It is difficult to sign labor contracts and obtain corresponding occupational benefits in the underemployed sector, and there is labor intensity, timing, and payment heterogeneity among different occupations [38]. Thus, there are differences in the impact of work characteristics on workers’ health. (5) In terms of participation in insurance, medical insurance can promote the participants’ health [39], pension insurance can provide basic livelihood for older adults [40], and unemployment protection expenditure can effectively improve the residents’ well-being [41]. In other words, participation in insurance can improve health outcomes.

## 3. Results

### 3.1. Descriptive Statistical Analysis

The descriptive statistics of the sample (Table 1) revealed that workers’ “self-rated health condition” was between “average“ and “healthy,“ and the mean values of “having a tendency toward depression” and “being sick for a period of time” were low. In the 2014 and 2016 sample data, the mean values of workers experiencing underemployment were 0.035 and 0.103, respectively. This indicates that the probability of experiencing underemployment in China is increasing. In addition, there was a minor difference in the percentage of men and women in the sample, and the majority of respondents had completed junior high school education. There were more workers in the secondary industry, but the percentages of those who work for others, those who were self-employed, and indoor workers and outdoor workers were close to the average. The percentages of workers who smoked or drank regularly were relatively low. However, we also found that few workers had unemployment insurance.

Table 2 reports health disparities among workers under different employment statuses, wherein the underemployed were at a disadvantage in terms of self-rated health, mental health, and prevalence of illness over a certain period of time compared to the fully employed and overemployed. The underemployed had the highest likelihood of poor self-rated health, depressive tendencies, and prevalence of illness over a certain period of time with 16.1%, 16.6%, and 36.1%, respectively. The health differences between the underemployed and fully employed and overemployed were significant at the 1% statistical level. This suggests an association between underemployment and workers’ health.

### 3.2. Short-Term Effects of Underemployment on Workers’ Health

The statistical association between underemployment and worker health could be explained by the following potential conditions: (1) underemployment affects health; (2) health affects the likelihood of workers being underemployed; and (3) other confounding variables affect the relationship between underemployment and worker health status. A simple comparison of means across the populations fails to control for the effects of other factors; consequently, all the core and control variables were used as explanatory variables. The health status of workers was used as the explained variable to estimate the sample data econometrically. To facilitate this analysis, the possible endogeneity of the sample was temporarily disregarded and the effect of underemployment on workers’ health was analyzed using a logit regression model.

The results in Table 3 reveal that the effect of underemployment on workers’ multidimensional health indicators passed the significance test at the 5% statistical level at the maximum level. That is, after controlling for workers’ individual differences, economic status, lifestyle, job characteristics, participation in insurance, and regional characteristics, odds of self-rated poor health, propensity to depression, and prevalence of illness over a certain period of time were 1.276, 1.328, and 1.338 times higher among the underemployed than the non-underemployed. The other factors were invariant. This suggests that the impact of underemployment on workers’ health is multidimensional in the short-term, with higher health risks compared to the non-underemployed. Further, underemployment status significantly impairs the overall health of workers.

The regression results revealed significant effects of several control variables on workers’ health. The results were largely consistent with previous studies. (1) In terms of individual characteristics, women’s health condition was worse than that of men; the health status of married workers was better, while that of divorced/widowed/cohabiting workers was “compromised,” possibly because marriage corresponds with a healthy diet that improves their health. The health status of workers was worse with age, and workers with better appearance had better health. (2) Both personal income and household income positively affected workers’ self-rated health, which is consistent with the findings of previous studies: residents with higher income tend to have better health conditions. (3) In terms of lifestyle, smoking positively affected workers’ self-rated health, which is contrary to the common perception. This could be explained by the fact that smokers do not consider smoking harmful to their health and believe that moderate alcohol consumption contributes to health [42]. Regular exercise positively affects workers’ self-rated health and mental health. (4) Regarding job characteristics, outdoor workers’ self-rated health was worse compared to indoor workers, and workers in the primary industry had worse self-rated health than those in the secondary and tertiary industries. This indicates that a better work environment could predict lower risk of injuries and disease [43]. (5) In terms of participation in insurance, having insurance reduced the workers’ health risk to some extent.

### 3.3. Endogenous Processing

#### 3.3.1. Propensity Score Matching

After matching workers for individual characteristics, economic status, lifestyle habits, job characteristics, and regional characteristics, Table 4 presents the differences and their significance levels for the treatment and control groups, respectively. For the full sample, the differences between the treatment and control groups reveal that, in terms of self-rated health, the underemployed had an average increase of 2.5% ((2.9% + 2.1%)/2) in health status compared to other workers. In terms of mental health, the underemployed had an average increase of 5.05% ((4.8% + 5.3%)/2) in the probability of displaying depressive tendencies compared to other workers. Finally, in terms of prevalence of illness over a certain period of time, the underemployed were on an average 5.75% ((5.8% + 5.7%)/2) more likely than other workers. Thus, although the effects of underemployment on workers’ self-rated health, mental health, and prevalence of illness over a certain period of time are lower after propensity matching than in the estimates based on logit regression, the results confirm that underemployment reduces workers’ overall health and that the presence of confounding variables objectively “amplifies” the effect of underemployment on workers’ health. Overall, underemployed workers are at a higher health risk than others.

#### 3.3.2. Instrumental Variable Method

The model estimation results (Table 5), revealed that underemployment reduces workers’ self-rated health status, increases the tendency of workers to be depressed, and increases the likelihood of prevalence of illness over a certain period of time compared to workers in other employment situations. The effect on all three factors is significant at statistical levels of 1%.

### 3.4. Robustness Test

To confirm the strong association between underemployment and workers’ health status, this paper examines the robustness of the findings by changing the sample (i.e., using CLDS 2014).

The estimation results based on CLDS 2014 data reveal that the effects of underemployment on workers’ health status passes the significance test at the 1% statistical level in all the cases (Table 6). This indicates that underemployment status significantly reduces the overall health status of workers. This also confirms that the previous findings are robust.

### 3.5. Long-Term Effects of Underemployment on Workers’ Self-Rated Health

Considering that the relationship between underemployment and workers’ health may change over time, we conducted further in-depth analysis using longitudinal data. Since CLDS data are tracking data, we matched and merged CLDS 2014 and CLDS 2016 data to analyze the impact of underemployment experience in 2014 on workers’ health status in 2016.

The estimation results in Table 7 reveal that after controlling for workers’ individual differences, economic status, lifestyle habits, job characteristics, participation in insurance, and regional characteristics, the probability of displaying depressive tendencies among the underemployed was 1.507 times higher than that of the non-underemployed, all else being equal. A worker’s experience of underemployment two years ago significantly increased present depressive tendencies. In other words, the negative effects of underemployment on workers’ mental health persisted over time. However, there was no significant long-term effect of underemployment experience two years ago on workers’ self-rated health and prevalence of illness over a certain period of time.

## 4. Discussion

As mentioned above, this paper argues that underemployment poses multiple health risks for workers. Although it can temporarily relieve workers from unemployment, it can impair health in the short-term and also affect long-term mental health status. Therefore, to reduce the damage caused by underemployment, more attention should be paid to this group of workers.

First, it is necessary to establish a database of the underemployed to protect their interests. Since this population (the hidden unemployed population) [44] is not included in the unemployed population by the government statistics department, the number of underemployed people and their health status in China has not attracted sufficient attention, which has resulted in insufficient research on the problem. Consequently, it has been difficult to formulate evidence-based and effective policy measures. Thus, it is suggested that the underemployed population should be considered as an important indicator when conducting unemployment rate statistics to adapt economic and social development.

Second, we must strengthen protective policies by considering health indicators for the formulation of employment policies and effectively promote the implementation of the “Health China” strategy. At the same time, we should improve the social security system and labor protection mechanism for underemployed workers, increase the effective supply of medical services, strengthen long-term psychological guidance, and improve the health level of underemployed workers. This must be done to promote the healthy and sustainable development of the labor supply.

However, the study has some limitations. Only two years of longitudinal data were used to explore the long-term effects of underemployment on workers’ health. This was because CLDS has published tracking data for three years and only a small sample was obtained after the data were matched with three years of information. Thus, only two years of tracking data were used for the analysis. However, we believe that future studies will gather more empirical evidence.

## 5. Conclusions

This paper analyzes the short- and long-term effects of underemployment on workers’ health status based on CLDS 2016 and CLDS 2014 data. The findings reveal that the impact of underemployment on workers’ health is multidimensional in the short-term, that is, underemployment is significantly associated with a decline in workers’ self-rated health in the current period, an increase in depressive tendencies, and an increase in prevalence of illness over a certain period of time. Furthermore, the experience of underemployment for two years in the past was significantly associated with a decline in workers’ current mental health. In other words, the negative effects of underemployment on workers’ mental health persisted over time.

This study is significant for two reasons. First, as the global spread of COVID-19 has negatively impacted the labor market, the phenomenon of underemployment is expected to become increasingly prominent. If workers are in a state of underemployment, it can adversely affect their physical and mental health in both the long- and short-term. Thus, this study provides a valuable reference for the formulation of future policies. Second, this study examines the health of workers from the perspective of underemployment in the Chinese context, which can contribute to the implementation of the “Healthy China” initiative, in particular, in the context of the fading traditional “demographic dividend.” Creating a new “demographic dividend” is important to promote economic development. Therefore, under the current situation of epidemic prevention and control, the government and society should pay more attention to this expanding group, establish labor protection mechanisms, and reduce the multiple effects of underemployment on workers’ health.

## Figures and Tables

**Figure 1 ijerph-19-16695-f001:**
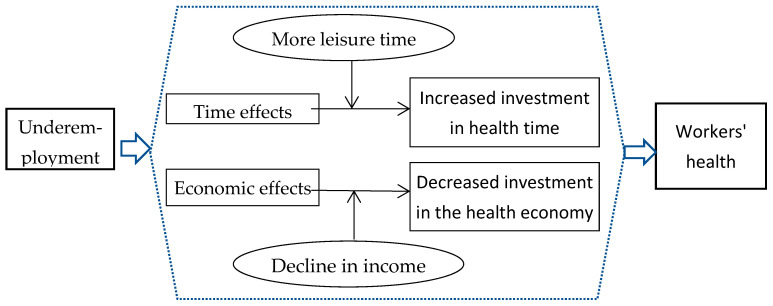
Theoretical framework.

**Table 1 ijerph-19-16695-t001:** Sample Characteristics.

Type	Variable Name		Variable Interpretation and Assignment	CLDS 2016		CLDS 2014	
				Average Value	Standard Deviation	Average Value	Standard Deviation
Dependent variable	Health Status	Self-assessment of health	1 = Very unhealthy; 2 = Rather unhealthy; 3 = Fair; 4 = Healthy; 5 = Very healthy	3.721	0.935	3.783	0.915
		Mental Health	1 = depressive tendency; 0 = no depressive tendency	0.153	0.360	0.456	0.498
		Prevalence of illness over a certain period of time	1 = Physical pain in the past month; 0 = No physical pain in the past month	0.304	0.460	0.285	0.452
Core Variables	Insufficient Employment		1 = underemployed; 0 = other	0.103	0.303	0.035	0.185
Control Variables	Individuals Features	Gender	1 = female; 0 = male	0.453	0.498	0.441	0.497
		Age	Actual age (years)	41.890	10.356	41.460	10.200
		Education level	1 = Elementary school and below; 2 = Junior high school; 3 = High School; 4 = College; 5 = Bachelor’s and above	2.269	1.173	2.295	1.199
		Appearance	1–10, 10 is the “highest” value	6.449	1.504	6.435	1.455
		Account	1 = non-farm household; 0 = agricultural household	0.258	0.437	0.281	0.449
		Marriage	1 = first marriage + remarriage; 0 = other	0.866	0.340	0.874	0.332
	Economy Status Features	Housing Sources	1 = owned property; 0 = other	0.510	0.500	0.841	0.366
		Household income	10,000 RMB	6.685	10.212	6.178	8.512
		Personal income	10,000 RMB	3.541	6.271	3.126	4.757
	Life Habit Features	Smoking	1 = yes; 0 = no	0.297	0.457	0.303	0.459
		Drinking	1 = drinks alcohol every day; 0 = no	0.075	0.263	0.230	0.421
		Regular exercise	1 = yes; 0 = no	0.287	0.452	0.197	0.398
	Jobs Features	Occupation Type	1 = employed by others; 0 = other	0.467	0.499	0.476	0.499
		Workforce Category	1 = transitory workforce; 0 = permanent employment	0.137	0.344	0.097	0.296
		Workplace	1 = indoor, workshop, office, home, collectively, and indoor; 0 = other	0.529	0.499	0.486	0.500
		Industry Properties	1 = primary sector; 2 = secondary sector; 3 = tertiary sector	1.998	0.883	1.968	0.871
	Enrollment Features	Medical Insurance	1 = yes; 0 = no	0.922	0.269	0.877	0.328
		Old-age insurance	1 = yes; 0 = no	0.655	0.475	0.585	0.493
		Unemployment Insurance	1 = yes; 0 = no	0.179	0.383	0.173	0.378
*N*				10,563		9703	

**Table 2 ijerph-19-16695-t002:** Health disparities among workers with different employment status.

Variables		Underemployment I	Full Employment II	Overemployment III	I VS. II	I VS. III	II VS. III
Self-assessment of health	1 = unhealthy, 0 = healthy	0.161 (0.007)	0.073 (0.005)	0.095 (0.004)	−0.088 ***	−0.066 ***	0.021 ***
Depressive tendencies	1 = yes, 0 = no	0.166 (0.004)	0.122 (0.006)	0.138 (0.012)	−0.044 ***	−0.028 **	0.016
Prevalence of illness over a certain period of time	1 = yes, 0 = no	0.361 (0.010)	0.255 (0.008)	0.302 (0.006)	−0.106 ***	−0.058 ***	0.047 **

Note: **, and *** indicate significance at the 5%, and 1% levels, respectively; a *t*-test was used to test the significance of the means of the two sets of variables.

**Table 3 ijerph-19-16695-t003:** Short-term effects of underemployment on workers’ health.

Variables	Self-Assessment of Health	Mental Health	Prevalence of Illness over a Certain Period of Time
Underemployment	0.244 ** (0.096)	0.284 *** (0.083)	0.291 *** (0.071)
Gender	0.120 (0.088)	0.443 *** (0.071)	0.428 *** (0.057)
Age	0.604 *** (0.044)	0.106 *** (0.033)	0.381 *** (0.027)
Education level	−0.191 *** (0.051)	−0.030 (0.036)	−0.074 ** (0.029)
Account	0.096 (0.126)	0.093 (0.089)	0.043 (0.072)
Appearance	−0.125 *** (0.024)	−0.099 *** (0.020)	−0.042 *** (0.016)
Marital Status	−0.097 (0.128)	−0.408 *** (0.086)	−0.181 ** (0.075)
Personal Income	−0.127 *** (0.021)	−0.007 (0.008)	−0.003 (0.005)
Household Income	−0.036 *** (0.010)	−0.007 (0.005)	−0.002 (0.003)
Housing	0.024 (0.071)	0.089 (0.058)	0.030 (0.047)
Smoking	−0.183 * (0.094)	0.076 (0.078)	−0.046 (0.061)
Drinking	−0.155 (0.125)	−0.065 (0.113)	−0.054 (0.086)
Regular Exercise	0.001 (0.089)	−0.178 *** (0.069)	−0.039 (0.054)
Occupation Type	−0.313 *** (0.114)	0.014 (0.082)	−0.171 *** (0.064)
Workplace	0.115 (0.109)	0.015 (0.080)	0.163 *** (0.063)
Industry Type	−0.203 *** (0.067)	−0.103 ** (0.050)	−0.088 ** (0.039)
Workforce Type	−0.026 (0.131)	0.187 ** (0.090)	0.016 (0.073)
Medical Insurance	−0.045 (0.132)	−0.091 (0.103)	−0.168 * (0.086)
Old-age Insurance	−0.054 (0.077)	−0.132 ** (0.064)	0.170 *** (0.052)
Unemployment Insurance	0.208 (0.159)	0.002 (0.102)	−0.062 (0.080)
Regional Fixed Effects	Controlled	Controlled	Controlled
Constant term	0.652 (0.487)	0.335 (0.344)	0.794 *** (0.286)
pseudo *R*^2^	0.157	0.035	0.070
*N*	10563	10563	10563

Note: *, **, *** indicate significance at the 10%, 5%, 1% levels respectively, with standard errors in parentheses.

**Table 4 ijerph-19-16695-t004:** Analysis of propensity score for underemployment and workers’ health status.

Health Status	Matching Method	Processing Group	Control Group	ATT Estimate	Bootstrap Standard Error	*T*-Value
Self-rated health (1 = unhealthy, 0 = healthy)	Nuclear matching	0.169	0.140	0.029	0.012	2.42 **
	K-Nearest Neighbor Matching	0.169	0.148	0.021	0.013	1.74 *
Mental health (1 = depressive tendency, 0 = no depressive tendency)	Nuclear matching	0.213	0.166	0.048	0.013	3.63 ***
	K-Nearest Neighbor Matching	0.213	0.160	0.053	0.017	3.07 ***
Prevalence of illness over a certain period of time (1 = sick, 0 = not sick)	Nuclear matching	0.398	0.340	0.058	0.016	3.65 ***
	K-Nearest Neighbor Matching	0.398	0.341	0.057	0.021	2.67 ***

Note: Considering the limitation of space, only the results of propensity score analysis are presented here, and the specific matching process has been prepared by the authors.* *p* < 0.10, ** *p* < 0.05, *** *p* < 0.01.

**Table 5 ijerph-19-16695-t005:** Underemployment and the short-term health status of workers: Instrumental variables.

Variables	Self-Assessment of Health	Mental Health	Prevalence of Illness over a Certain Period of Time
Underemployment	3.193 *** (0.186)	2.847 *** (0.681)	2.690 *** (0.520)
Control variables	Controlled	Controlled	Controlled
Regional fixed effects	Controlled	Controlled	Controlled
Constant term	0.360 (0.239)	−1.000 *** (0.117)	−1.099 *** (0.108)
*N*	10,563	10,563	10,430

Note: *** indicate significance at the 1% level, respectively, with standard errors in parentheses.

**Table 6 ijerph-19-16695-t006:** Underemployment and workers’ short-term health status: Robustness tests.

Variables	Self-Assessment of Health	Mental Health	Prevalence of Illness over a Certain Period of Time
Underemployment	0.783 *** (0.154)	0.789 *** (0.118)	0.423 *** (0.121)
Control variables	Controlled	Controlled	Controlled
Regional fixed effects	Controlled	Controlled	Controlled
pseudo *R*^2^	0.153	0.039	0.075
*N*	9703	9703	9703

Note: *** indicate significance at the 1% level, respectively, with standard errors in parentheses.

**Table 7 ijerph-19-16695-t007:** Long-term effects of underemployment on workers’ health.

Variables	Self-Assessment of Health	Mental Health	Prevalence of Illness over a Certain Period of Time
Underemployment	−0.186 (0.272)	0.410 * (0.213)	−0.134 (0.195)
Control variables	Controlled	Controlled	Controlled
Regional fixed effects	Controlled	Controlled	Controlled
pseudo *R*^2^	0.159	0.045	0.073
Sample size	4713	4713	4713

Note: * indicate significance at the 10% level, respectively, with standard errors in parentheses.

## Data Availability

The data from the China Labor-force Dynamics Survey used in this study are available from the Centre for Social Survey, Sun Yat-sen University. Requests to access the datasets should be directed to the authors, and access will be granted upon approval from CSS.

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
