# Peer review of "Short- and Long-Term Effects of Underemployment on Workers’ Health: Empirical Analysis from the China Labor Force Dynamics Survey"

_ijerph, 2022, doi:10.3390/ijerph192416695_

Round 1

Reviewer 1 Report

I want to thank the authors for their submission on the effects of underemployment on individual health. The idea of the article is interesting and the approach to use an existing dataset to not only test the relationships, but also check their robustness is sound. There are mainly minor details and questions that I would like the authors to address in order to further elevate the quality of the article. I want to discuss these issues in the order that they appear in the manuscript and I will separate them into further methodological details that are needed and some minor issues.

Methodological details. On p. 3: The authors should provide some additional information on the two samples and how data collection for the CLDS usually works (e.g., types of data collection involved, representativeness of the sample for the Chinese population) in order for the reader to have a clearer picture of the dataset and its utility for this study. On p. 3: The authors should also include the information that 9703 individuals were included from the sample of the CLDS 2014 (as indicated in Table 6). On p. 3: The authors should include references on the utility / methodology of Propensity Score Matching and the instrumental variables method. On p. 4: Why was self-rated health transformed from five levels into two? This transformation creates a loss in information, while it is possible to also include outcome variables with more than two categories (multinomial logistic regression instead of binary logistic regression). On p. 6, Table 1: For “Appearance” it should be added that 10 is the “highest” value. For the two income features the authors should check again if the data here is correct. Is 6 million+ yuan the lifetime income or the yearly income? If it is the yearly income, then it would be way above the average for China at around 90.000 to 100.000 yuan (https://www.tradecommissioner.gc.ca/china-chine/wages-salaires.aspx?lang=eng). On p. 10, Table 4: Is it possible that the values for Self-rated health were mixed up? Based on the description in the text, the values for the Processing group should be higher (i.e., .574 and .567) than the values for the Control group (i.e., .517 and .517), but currently it is the other way around.

 Minor issues (examples for typesetting issues are exemplary; potential remedies in brackets):

 p. 2, line 48: “…workers' health.[ ]Mushkin (1962) proposed that "health is an investment[]"…”

p. 2, line 75: “…combined []time and economic effects.”

p. 4, line 196: there is missing text at “…desired (underemployment) …the average weekly workings hours…”

The third well-being variable “occurrence of an illness” was renamed frequently, for example as “A certain period illness” in Table 1 and “A certain period is the illness” in Table 2, or “Prevalence over a certain period of time” in Table 3. The used label should be kept consistent.

p. 8, line 265: The authors used the term “explanatory variables” both for dependent and independent variables

p. 9 and p. 12: there is no need to express multitudes such as 1.276 also with an exponential (e0.244). Such redundant information should be avoided.

p. 12: When the authors mention the “hidden unemployed population” (line 370) a reference should be added to further explain the origin of this label.

Overall a nice piece of research that is almost ready for publication. Independent of the outcome of the review process, I wish the authors all the best for their future research.

Author Response

Dear Reviewer:

Thank you very much for your careful review and constructive comments on the thesis. According to your specific comments, the manuscript has been revised and improved, and the specific changes are described as follows.

Point 1:. On p. 3: The authors should provide some additional information on the two samples and how data collection for the CLDS usually works (e.g., types of data collection involved, representativeness of the sample for the Chinese population) in order for the reader to have a clearer picture of the dataset and its utility for this study.

Response 1: Thank you very much for your review. We have detailed CLDS in the resubmitted manuscript.

CLDS is a tracking database, which integrates labor force, households, and communities. Probability sampling method was used at multiple stages and levels proportional to the size of the labor force to determine the sample. This database is the first in China to adopt a rotating sample tracking method, which suitably adapts to the drastic changes in China while accounting for the characteristics of the cross-sectional surveys. The sample covers 29 provinces, cities, and autonomous regions in China, except Hong Kong, Macau, Taiwan, Tibet, and Hainan, and is nationally representative.

Point 2: On p.3: The authors should also include the information that 9703 individuals were included from the sample of the CLDS 2014 (as indicated in Table 6).

Response 2: Thank you very much for your review. We agree with your opinion. We have added “the CLDS 2014 retained 9703 observations” to the manuscript.

Point 3: On p. 3: The authors should include references on the utility / methodology of Propensity Score Matching and the instrumental variables method.

Response 3: Thank you very much for your review. We agree with your opinion. We have added references on the utility/methodology of Propensity Score Matching and the instrumental variables method in the manuscript.

  1. [20] Becker, S.O.; Ichino, A. Estimation of average treatment effects based on propensity scores. Stata J2002, (2), 358– doi:10.1177/1536867X0200200403.
  2. [21] Chen, Q. Advanced econometrics and stata application. Higher Education Press, 2014.

Point 4:On p. 4: Why was self-rated health transformed from five levels into two? This transformation creates a loss in information, while it is possible to also include outcome variables with more than two categories (multinomial logistic regression instead of binary logistic regression).

Response 4: Thank you for your review. In the analysis, we transformed the self-rated health from five levels to two based on two considerations. First, in the analysis of health differences among workers with different employment status, a t-test (t test) was used to dichotomize self-rated health. Secondly, in the empirical analysis, since both mental health and prevalence over a certain period of time are dichotomous variables, self-rated health was also determined as a dichotomous variable to maintain model’s consistency. Moreover, this setting did not change the validity of the statistical results.

The results of the logit statistics when self-rated health is a five-category variable are shown in the table below. The results revealed that underemployment significantly reduced workers' self-rated health, which is consistent with the following result: self-rated health is a dichotomous variable.

Variables

Self-assessment of health

Underemployment

-0.262***

(0.062)

Point 5.On p. 6, Table 1: For “Appearance” it should be added that 10 is the “highest” value. For the two income features the authors should check again if the data here is correct. Is 6 million+ yuan the lifetime income or the yearly income? If it is the yearly income, then it would be way above the average for China at around 90.000 to 100.000 yuan.

Response 5: Thank you for your review. We have added that 10 is the highest value for appearance in the manuscript. In terms of household income and personal income, the units are RMB 10,000, which were incorrectly included in the initial manuscript. I am particularly grateful to the reviewer for pointing out this problem..

Point 6. On p. 10, Table 4: Is it possible that the values for Self-rated health were mixed up? Based on the description in the text, the values for the Processing group should be higher (i.e., .574 and .567) than the values for the Control group (i.e., .517 and .517), but currently it is the other way around.

Response 6: Thank you for highlighting the problem. There was indeed an obvious error in this section, which has been meticulously re-analyzed. All econometric models throughout the text have been re-checked and proofread in accordance with the expert's comments.

Point7:p. 2, line 48: “…workers' health.[ ]Mushkin proposed that "health is an investment[]"…”

Response 7: Thank you for your comment. Here are two natural paragraphs, which have been subdivided in the resubmission of the manuscript.  “…workers' health” are the last sentence in the first paragraph, and “Mushkin proposed that "health is an investment"…” are the first sentence in the second paragraph.

Point 8:p. 2, line 75: “…combined []time and economic effects.”

Response 8: Thank you for your comment. Here, “time effects and economic effects” was missing and has been corrected in the resubmitted manuscript.

Point 9:p. 4, line 196: there is missing text at “…desired (underemployment) …the average weekly workings hours…”

Response 9: Thank you for pointing out the problem. We have corrected the sentence as follows: “…desired (underemployment). Therefore, referring to previous studies, this paper defines the average weekly workings hours…”. 

Point 10:The third well-being variable “occurrence of an illness” was renamed frequently, for example as “A certain period illness” in Table 1 and “A certain period is the illness” in Table 2, or “Prevalence over a certain period of time” in Table 3. The used label should be kept consistent.

Response 10: Thank you for pointing out the problem. We have re-read the entire text and harmonized the “occurrence of an illness” to “prevalence of illness over a certain period of time.”

Point 11:p. 8, line 265: The authors used the term “explanatory variables” both for dependent and independent variables

Response 11: Thank you for pointing out the problem. We have revised this in the manuscript. The core variables and all control variables are used as explanatory variables, and the health status of workers is used as the explained variable.

Point 12:p. 9 and p. 12: there is no need to express multitudes such as 1.276 also with an exponential (e0.244). Such redundant information should be avoided.

Response 8: Thank you for your review. We have removed these exponentials in the resubmitted manuscript.

Point 13:p. 12: When the authors mention the “hidden unemployed population” (line 370) a reference should be added to further explain the origin of this label.

Response 8: Thank you for your comment. We have added references in the resubmitted manuscript.

  1. [37] Wang, Q.F.; Tetiana. L. Immigrant underemployment across US metropolitan areas: From a spatial perspective . Urban Stud 2014, 51(10), 2202– doi:10.1177/0042098013506043.
  2. [38] Katina, W.; Thompson, T.H.; et al. Rethinking underemployment and overqualification in organizations: The not so ugly truth.Bus Horiz 2013, 56, 113– doi:10.1016/j.bushor.2012.09.009.

Reviewer 2 Report

Dear authors,

thank you for giving me the chance to review your manuscript on "Short- and long-term effects of underemployment on workers' health: Empirical analysis from the China Labor Force Dynamics Survey".

Not only with regard to the COVID-19 pandemic, this topic may shed light on important health risks of employees. 

However, below, I want to share some remark that should be addressed to make the manuscript stronger:

Abstract:

- Not clear why anybody from outside China should be interested in the research.

- Text content is too broad and without focus. 

Introduction:

- It is unclear what is the background and motivation of the article.

- How did the authors come to the research gap? It looks like they just "invented" it themselves.

- Theoretical framework is not present. There are not enough references and those are very poorly linked to the few models mentioned.

Methodology: 

- While the authors acknowledge correctly that the data are old - newer data does not exist. The authors should explain why should the world be interested in 6-8 year old data from China in that case.

- The authors do not disclose information the intitial survey and whether the amount of data used is representative & allows for generalization. 

Results:

- The authors should clarify in the manusript text whether they are talking about "health" or "mental health" as an outcome of underemployment. What does the "self-assessment of health" refer to?

- I assume "underemployment" also includes "unemployment" in the tables? If so, the analysis has no substance & has to be re-done.  

Formal:

- Abstract form

- Text should be set in block,

- References not according to guidelines,

- Big tables should have their place in the appendix,

- Tables should be better formatted, as they are hard to read,

- Please check your English - there are several mistakes, double commas, etc.  

I wish the authors all the best

Best regards

Author Response

Dear Reviewer:

Thank you very much for your careful review and constructive comments on the thesis. According to your specific comments, the manuscript has been revised and improved, and the specific changes are described as follows.

Point 1. Abstract:

Point 1.1- Not clear why anybody from outside China should be interested in the research.

Response 1.1: Thank you very much for your review. We have modified the abstract in the manuscript. In order to explain why anybody from outside China would be interested in the research, we have reviewed a large amount of literature to improve the following testimonial material.

Underemployment is a global problem, particularly as economic recessions and radical changes in the labor market have caused the collapse of stable jobs, which has resulted in a large proportion of the workforce becoming underemployed (International Labour Organization, 2017). However, statistics from labor force surveys around the world show that attention is often focused on those workers who are unemployed and neglects those who are underemployed. Usually, the unemployment rate does not reflect the under-utilization of a country's human resources, and if underemployment is not taken into account, the unemployment rate will be greatly underestimated, which hampers the certainty of our conclusions. As underemployment involves a large number of workers and, a wide range of geographical, economic, and demographic conditions, it is necessary to raise awareness and understanding of underemployment. It is important to conduct research on the impact of underemployment on workers.

Point 1.2- Text content is too broad and without focus.

Response 1.2:Thank you for your comment. We have modified the abstract to make the content clearer.

Objective: Underemployment is a global problem, the purpose of this study was to assess the short- and long-term effects of underemployment (hidden unemployment) on workers' health. Methods: Latest available data from the China Labor-force Dynamic Survey (CLDS) 2016 and 2014 and indicators reflecting workers' self-rated health, mental health, and prevalence of illness over time, and employment status were used as data. The data were subjected to Logit regression models, propensity score matching methods, and instrumental variable methods. Results: Empirical analysis showed the following: (1) in the short-term, the impact on health is multidimensional, with underemployment significantly associated with a decline in workers' self-rated health, an increase in the propensity for depression, and an increase in the prevalence of illness over a certain period of time. (2) In the long-term, the experience of underemployment two years in the past is associated with a current decline in workers' mental health. That is, the negative effects of underemployment on workers' mental health persist and do not disappear rapidly over time. Conclusions: The results demonstrated that underemployment is detrimental to workers' health in the short- and long-term. In the context of epidemic prevention and control, the government and society should focus on this expanding group, establish labor protection mechanisms, and reduce the multiple effects of underemployment on workers' health.

Point 2. Introduction:

Point 2.1- It is unclear what is the background and motivation of the article.

Response 2.1:Thank you for your review. We have modified the introduction in the manuscript.

Underemployment is a global problem, in many countries, growth in employment is an important cause of economic growth. But the increase in jobs does not mean an increase in the quality of work. In reality, there is an increase in informal work and more workers find themselves in jobs that do not pay enough to lift them and their families out of poverty (Bell and Blanchflower,2019). So, in many countries, it is not unemployment that is the problem, but rather the existence of a large number of “underemployed” with no prospects for development. But statistics from labor force surveys around the world show that the focus of attention is often on those who are unemployed, while neglecting those who are underemployed. Specifically in China, the neglect of the underemployed may be due to three reasons: Firstly, with the gradual receding of the demographic dividend, there is an oversupply of labor in the labour market, and the phenomenon of "difficulty in recruiting" for enterprises has gradually come to the fore, as if there is no underemployment problem; secondly, the officially announced urban registered unemployment rate has remained stable at a low level (around 4%) for a long time. Thirdly, under the exaggeration of media reports, overemployment, overtime work and overwork have flooded the whole labor market, making people mistakenly believe that only the problem of overemployment exists, while neglecting to pay attention to underemployment. Therefore, understanding the issue of underemployment not only helps to correctly judge the extent of underutilization of labor resources, but also alleviates to a certain extent the contradiction between people's growing aspiration for a better life and underemployment.

Health is the common well-being of human beings, and among the many factors affecting individual health, social factors are considered to be the main health factors affected as opposed to medical-technical factors, and in studies on health inequality, an individual's socioeconomic status is considered to be the most fundamental antecedent variable (Zhu and Feng, 2018), and as the measurement of socioeconomic status is mostly closely linked to an individual's employment status, the As a form of hidden unemployment, underemployment largely determines the socioeconomic status of workers, which in turn can lead to the occurrence and development of health inequalities.

Point 2.2- How did the authors come to the research gap? It looks like they just "invented" it themselves.

Response 2.2:Thank you for your comment. Please review the following information:

While several studies have examined the effects of unemployment and overwork on workers' health, only a few have assessed the relationship between underemployment and workers' mental health using cross-sectional data. In fact, there are even fewer studies that have confirmed the negative effect of underemployment on the mental health of individuals [12,13]. Although many scholars have emphasized the importance of employment quality, the issue remains controversial. For example, Jahoda suggested that even a bad job is better than unemployment and argued that there is no association between underemployment and the health of workers [14]. However, Wu suggested that short-time jobs, though technically not underemployment, adversely affect health because they do not meet the psychosocial and economic needs of workers [15]. Underemployment can be described as a potential social stressor that pressurize the workers and may endanger their health. According to the Effort-Reward Imbalance (ERI) model, work can lead to poor mental health if people are not appropriately "satisfied" or "rewarded" [16]. Evidence also suggests that any change in role or the environment that objectively requires adaptation, causes a specific stress response which accumulates over time and causes varying degrees of damage to health [17,18]. Every employee is exposed to different social stressors but they cope with the stress differently. For example, a study on East Asian immigrants and Vancouver residents found that underemployment was associated with higher levels of depression among Vancouver residents, while the same was not true for East Asian immigrants [19]. Thus, underemployment affects workers' health, but the effects are neither as strong nor as consistent as is commonly believed.

In summary, there are several research lacunas to be filled. First, most studies on employment status and health simply distinguish between employment and unemployment without considering underemployment. This conceals the complexity of the relationship between work and health. Second, most studies on the impact of underemployment on workers' health have focused on a single disease or health indicator and lack multidimensional in-depth analysis. This may greatly underestimate or overestimate the impact of underemployment on health. Therefore, this study discusses the relationship between underemployment and a series of physical and mental health indicators. Third, since existing studies lack longitudinal analysis, this paper uses tracking data to analyze whether underemployment affects workers' health in the long-term.

Point 2.3 - Theoretical framework is not present. There are not enough references and those are very poorly linked to the few models mentioned.

Response 2.3:Thank you for your review. We apologize for the lack of theoretical information in the manuscript. We have included the following in the manuscript to resolve the issue. Mushkin proposed that "health is an investment " which determines the amount of time people spend on market or non-market productive activities, and their effectiveness per unit of time [6]. It has been suggested that, given a certain level of socioeconomic development and other external conditions, an individual's investment in health largely influences their health condition. The investment in health mainly comprises two aspects: economic investment (e.g., improving quality of life) and time investment (e.g., engaging in physical activity). Since each day has a fixed number of hours, the time a person invests on health and work are necessarily negatively related, provided that other conditions are invariant. Since underemployed people spend less time working, they have relatively more time to invest on health and for physical activities [7]. Therefore, they are likely to be in better health condition.

Meanwhile, Grossman argued that "health is a consumer goods" which frees people from the suffering caused by disease or illness, and gives them a sense of satisfaction and utility. He also proposed the first theoretical model for analyzing the demand for health. The model argues that people can purchase health in a monetary or nonmonetary manner [8]. Since underemployment is a kind of hidden unemployment, long-term under-employed workers usually do not have guaranteed income, and their health care ex-penditure and other general expenditure is constrained by their income which hinders their overall health condition [9,10]. Moreover, low income can also cause psychological burden or stress [11], which increases the depreciation rate of their health capital. Therefore, under the given external conditions, the incremental health capital of the un-deremployed is lower than that of fully employed workers.

Since underemployment is associated with more leisure time and possibly lower income, this paper argues that the time effect caused by underemployment has a positive impact on workers' health, while the economic effect has a negative impact on workers’ health. The overall effect of underemployment is the combination of its time and economic effects.

Due to space constraints, the results of the impact of underemployment on workers' health are not represented in the paper. The three possible outcomes of the impact of underemployment on workers' health are detailed here using the health equilibrium output diagram.

  • Underemployment is beneficial forworkers' health if the positive effect of time investment is greater than the negative effect of economic investment on workers' health. As shown in Figure 1, underemployment leads to an increase in leisure time (non-working time), which leads to an increase in the amount of time workers devote on health. This in turn, has a positive effect on health. At this juncture, the health marginal benefit curve shifts from MR1 to MR3 to the right, and the equilibrium health status of workers rises from h1 to h3. Further, underemployment may also cause a decrease in the income level of workers, which leads to a decrease in the economic investment in health. At this juncture, the health marginal benefit curve shifts from MR3 to MR2 to the left again, and the equilibrium health status of workers falls from h3 to h2. The positive effect of time investment on workers' health from full employment is greater than the negative effect of economic investment on workers' health. Overall, workers' equilibrium health status increases from h1 to h2, and underemployment is beneficial for workers' health.

Figure 1 Healthy and balanced output1

  • The positive effect of time investment onworkers' health due to underemployment is equal to the negative effect of economic investment to workers' health, then underemployment has no effect on workers' health. As shown in Figure 2, underemployment causes an increase in workers' time investment in health. This has a positive impact on health. At this juncture the health marginal benefit curve shifts from MR1 to MR3 to the right, and workers' equilibrium health rises from h1 to h3. At the same time, economic investment brought about by underemployment has a negative impact on health, at which time the health marginal benefit curve shifts from MR3 to the left again to MR2, and the equilibrium health condition of workers decreases from h3 to h2. At this juncture, the positive impact of time investment brought by underemployment on workers' health is equal to the negative impact of economic investment on workers' health.

Figure 2 Healthy and balanced output2

  • The positive effect of time investmentdue to underemployment on workers' health is smaller than the negative effect of economic investment on workers' health. In this case, underemployment is detrimental to workers' health. As shown in Figure 3, the time investment brought by underemployment shifts the marginal return curve of health MR1 to the right to MR3, and the equilibrium health condition of workers rises from h1 to h3. Meanwhile, the economic investment brought by underemployment shifts the marginal return curve of health from MR3 to the left to MR2, and the equilibrium health condition of workers falls from h3 to h2. At this time juncture, the positive effect of time investment on workers' health is smaller than the negative effect of economic investment on workers' health. Overall, the equilibrium health status of workers decreases from h1 to h2, and underemployment is detrimental to workers' health.

Figure 3 Healthy and balanced output3

Point 3. Methodology

 Point 3.1- While the authors acknowledge correctly that the data are old - newer data does not exist. The authors should explain why should the world be interested in 6-8 year old data from China in that case.

Response 3.1:Thank you for your review. As there is still a lack of recent micro data on the current epidemic, the China Labor-force Dynamics Survey (CLDS) 2016 and 2014 data, publicly available from the Social Science Research Centre of Sun Yat-sen University, are used for the econometric analysis. Although the data itself is not for the epidemic period, the findings of this paper may can be used as reference for studying the impact of underemployment on workers' well-being under the epidemic. This is because stable employment has become increasingly constrained by the impact of the epidemic, and underemployment remains continuing problem. Given this economic environment, particular attention must be paid to the health consequences of underemployment. In addition, we look forward to the release of updated data.

Point 3.2- The authors do not disclose information the initial survey and whether the amount of data used is representative & allows for generalization.

Response 3.2: Thank you for your comment. We have detailed CLDS in the resubmitted manuscript.

CLDS is a tracking database, which integrates labor force, households, and communities. Probability sampling method was used at multiple stages and levels proportional to the size of the labor force to determine the sample. This database is the first in China to adopt a rotating sample tracking method, which suitably adapts to the drastic changes in China while accounting for the characteristics of the cross-sectional surveys. The sample covers 29 provinces, cities, and autonomous regions in China, except Hong Kong, Macau, Taiwan, Tibet, and Hainan, and is nationally representative.

Point 4. Results:

Point 4.1- The authors should clarify in the manuscript text whether they are talking about "health" or "mental health" as an outcome of underemployment. What does the "self-assessment of health" refer to?

Response 4.1: Thank you for your review. The health indicators used in this paper are from a three-dimensional health assessment system that includes workers' self-rated health, psychological health, and prevalence of illness over time. In the short term, underemployment reduces the overall health of workers. In the long-term, the negative effects of underemployment persist. This suggests that the most serious impact of underemployment on workers is at the psychological level, that is, the effects of underemployment on workers' self-rated health and prevalence of illness over time disappear, while the negative impact on mental health persists.

"self-assessment of health" is an individual's comprehensive feeling and evaluation of various aspects of their health status, and is closely related to the work stress of workers. In the CLDS questionnaire, self-rated health is an ordered discrete variable corresponding to the question, "How healthy do you think you are now?" , the options for this question include "very healthy, healthy, fair, relatively unhealthy, very unhealthy".

Point 4.2- I assume "underemployment" also includes "unemployment" in the tables? If so, the analysis has no substance & has to be re-done.

Response 4.2: Thank you for your review. The workers analyzed in this paper are the underemployed population and do not include the unemployed. The underemployed are now more normalized and deemed as individuals who work less than 35 hours/week and wish to work longer. A detailed description of the core explanatory variables of this paper is given on page 5 of the paper.

References

Bell, D. N. F.; Blanchflower, D. G. Underemployment in the United States and Europe. ILR Review 2019, 74(1), 56–94. doi:10.1177/0019793919886527

Kim, T.; Allan, B.A. Underemployment and meaningful work: The role of psychological needs. J Career Assess 2020, 28(1):76-90.

International Labor Organization. World employment and social outlook: Trends 2017. Geneva, Switzerland, 2017.

Zhu, H.J.; Feng, X.T. Health inequalities in the context of “Healthy China”. Learning Pact 2018, (4), 91–98.

Round 2

Reviewer 2 Report

Dear authors,

thank you for giving me the chance to review your manuscript on "Short- and long-term effects of underemployment on workers' health: Empirical analysis from the China Labor Force Dynamics Survey".

Not only with regard to the COVID-19 pandemic, this topic may shed light on important health risks of employees. 

However, below, I want to share some remark that should be addressed to make the manuscript stronger:

Introduction:

- As the authors try to explain underemployment negatively affects productivity: There are ample studies to show that less daily work time may result in higher or same productivity. How did the authors evaluate these studies in their context?

- The research gaps should be fed with several, previously-found gaps. Gap 2 & 3 are only supported by one source.

Literature review:

- The theoretical framework should be drawn at the end of the chapter. Five sources is not enough & crucial further factors seem not not be taken into account.

Methodology: 

- The authors should explain in the manuscript why there is no newer data available & why the authors estimate the used data to be suitable today.

- The authors should reveal more information on the CLSD & the questions asked in it.  

Results:

- The authors should clearly explain that “self-assessed health“ includes “mental health“ and “physical health“

- The authors should make clear that the results for “underemployment“ do not include unemployment.

Formal:

- Abstract should be unstructured,

- Mix of citation styles in the article,

- Big tables should have their place in the appendix,

- Low number of references,

- Tables should be better formatted, as they are hard to read,

- Please check your English - there are several mistakes, double commas, etc.  

I wish the authors all the best

Best regards

Author Response

Dear Reviewer:

Thank you very much for your careful review and constructive comments on the thesis. According to your specific comments, the manuscript has been revised and improved, and the specific changes are described as follows. All revisions to the manuscript have been marked up using the “Track Changes”, and use yellow highlight to indicate new content in the manuscript.

Point 1. Introduction

Point1.1- As the authors try to explain underemployment negatively affects productivity: There are ample studies to show that less daily work time may result in higher or same productivity. How did the authors evaluate these studies in their context?

Response 1.1: Thank you very much for your review. Although less daily work time may result in higher or same productivity, for the vast majority of people, obtaining a source of income through employment is a basic and primary means of survival. In this context, underemployment is not really the 'ideal job' that workers aspire to, but rather an inferior job that they have no choice but to do. Although, as income levels rise and social welfare as a whole improves, people generally prefer to work at leisure for more satisfaction, the number of workers who work short hours voluntarily is, after all, a minority and the majority of those who work short hours do so involuntarily. As reducing the working hours of employees is one of the strategies used by employers to cope with reduced demand for services and products in times of general economic uncertainty, underemployment allows for 'job sharing' in the labor market by reducing the working hours of workers. It not only helps companies (on the demand side) to control labor costs, increase total employment, reduce the incidence of unemployment, and also meet the diverse employment needs of job seekers.

Point 1.2- The research gaps should be fed with several, previously-found gaps. Gap 2 & 3 are only supported by one source.

Response 1.2:Thank you very much for your review. We agree with the opinions of the reviewers. We have combined Gap 2 and Gap 3 (in the manuscript, 1.2.Underemployment and health: A literature review, P3).

Point2. Literature review:

The theoretical framework should be drawn at the end of the chapter. Five sources is not enough & crucial further factors seem not not be taken into account.

Response 2:Thank you very much for your review. We have adapted the theoretical framework to fit the end of the chapter and have added more literature review to our manuscript (in the manuscript, 1.1. Health as a commodity: A theoretical analysis, P2). 

Point 3.Methodology

 Point3.1- The authors should explain in the manuscript why there is no newer data available & why the authors estimate the used data to be suitable today..

Response 3.1:Thank you very much for your review. We have explained in the manuscript why there is no newer data available and why the authors estimate the data used to be suitable today( in the manuscript, 2.1. Data sources, P4).

Point3.2- The authors should reveal more information on the CLSD & the questions asked in it.

Response 3.2: Thank you very much for your review. We have detailed the CLDS, including the questions asked in it, in the resubmitted manuscript(2.2.1. Dependent variables, P5).

Point 4.Results:

Point4.1- The authors should clearly explain that “self-assessed health” includes “mental health” and “physical health”

Response 4.1: Thank you very much for your review. Self-rated health is one of the most important subjective indicators of population health and an important basis used by many scholars to measure the health of the population, probably because self-rated health is relatively easy to obtain, high quality variable in surveys. Self-rated health is a subjective assessment of a respondent's health based on comparisons of their own health status with others around them. In the manuscript (2.2.1. Dependent variables, P5) we have explained self-rated health in detail.

Point4.2- The authors should make clear that the results for “underemployment“ do not include unemployment.

Response 4.2: Thank you very much for your review. In the manuscript (2.2.2. Core variables, P5) we have explained the definition and measurement of underemployment in detail. Specifically, an average working week of between 0 and 35 hours, and dissatisfaction with the number of working hours is defined as underemployment. We could see that the extracted data for underemployment do not include unemployment, and therefore believe that they do not influence the results as well.

Formal:

- Abstract should be unstructured,

Thank you very much for your review. The abstract has been made unstructured, formatted, and edited to within the word limit of 200.

- Mix of citation styles in the article,

Thank you very much for your review. The manuscript has been carefully checked for any deviations from the prescribed citation style.

- Big tables should have their place in the appendix,

Thank you very much for your review. We have gone through the template and the website , it says tables that are important, but not essential to the article flow can be placed in the appendix. In our manuscript we think that big tables (table 1 and table 3) are essential to the article flow, so they are not in the appendix. If you think they should be shown in the appendix, we would be happy to move the two tables to the appendix.

- Low number of references,

Thank you very much for your review. Additional relevant references have been added to strengthen the study in our manuscript.

- Tables should be better formatted, as they are hard to read,

Thank you very much for your review. Tables have been reformatted such that they are easier to read.

- Please check your English - there are several mistakes, double commas, etc.

Thank you very much for your review. The paper has been professionally edited by a native English language speaker, and reviewed for any errors in language.
